# Depth Partitioning and Diel Movement of Two Large Carcharhinid Sharks in Extremely Shallow Waters

Adi Barash [1,2,*], Aviad Scheinin [1,3], Eyal Bigal [1,3], Ziv Zemah Shamir [1,3], Stephane Martinez [1,3] and Dan Tchernov [1,3]

1   Leon Charney School of Marine Sciences, University of Haifa, Haifa 3498838, Israel
2   Sharks in Israel, NGO for the Conservation of Sharks and Rays, Kibbutz Amir 1214000, Israel
3   Morris Kahn Marine Research Station, University of Haifa, Haifa 3498838, Israel
*   Correspondence: adibarash@hotmail.com

**Abstract:** Two species of carcharhinid sharks aggregate every winter at the warm water effluent of a coastal power plant on the Israeli Mediterranean coast. The two species (*Carcharhinus obscurus* and *Carcharhinus plumbeus*) cooccur in a highly confined area for several months every year and are highly associated with the area in and around the hot water effluent. Niche partitioning has recently been suggested as a mechanism that enables the coexistence of similar shark species by resource partitioning, spatial partitioning, and temporal partitioning. In this study, we used acoustic telemetry to study the individual diel movement and activity patterns within this enclosed area and examined the differences between the two species sharing it. Although this location only reaches a maximum depth of 7.5 m, we found both species perform a diel vertical movement, rising closer to the surface at night and moving deeper during daytime. Furthermore, the two shark species swam at different depths both day and night, with *C. obscurus* swimming in the upper column, about 2 m shallower than *C. plumbeus*. The very small scale of movement, which nearly equals the sharks' body length, suggests movement patterns might be conserved at the species level. Moreover, spatiotemporal differences between the two species may reflect a mean of interspecific partitioning that occurs even in a highly confined and shallow habitat.

**Keywords:** partitioning; spatial patterns; predators; selacii; elasmobranch; habitat selection; *Carcharhinus obscurus*; *Carcharhinus plumbeus*; competition; behavioural plasticity

## 1. Introduction

In the Eastern Mediterranean, carcharhinid sharks aggregate near the coast of Israel at warm water effluents of coastal power stations [1]. Every year for the last two decades, dozens of sharks of two species, the dusky shark *Carcharhinus obscurus* (Lesueur, 1818) and the sandbar shark *C. plumbeus* (Nardo, 1827), aggregate in this relatively small area between November and May, most likely due to elevated temperatures and their thermoregulatory advantages [1]. Both species are large coastal sharks (*C. obscurus* up to 4.2 m and *C. plumbeus* up to 2.5 m [2]), with similar food preferences and trophic levels [3], and seem to coexist in large numbers in a small and extremely shallow area.

Niche partitioning has been found to be a significant mechanism allowing multiple species to share common space or resources [4]. Studies have shown that in areas where different species of large sharks coexist, differences were found in the use of space among species. For example, in Queensland, Australia, two shark species inhabit close but separated areas along the same river [5]. Around a small, elongated island near Mexico, four species of sharks have been documented with high affinity to only one site on the island, suggesting spatial partitioning for some of the species [6]. Six shark species in the Gulf of Mexico showed a diel temporal partition when each species utilized the same space at a different time of the day, with minimal overlap between the activity hours [7].

Little is known about niche partitioning in terms of depth distribution. Based on isotope analysis of mercury accumulation, reference [8] suggested that foraging depth can explain resource allocation between species, and reference [6] described the varied use of depth among individuals of different species on the same site.

In this study, we used acoustic telemetry to examine how two large coastal shark species coexist within a small area of a few kilometres, limited by extremely shallow water. We also examine the hypothesis that niche partitioning facilitates their coexistence.

## 2. Materials and Methods

### 2.1. Study Site

Orot Rabin (OR) power station (32.466814 N, 34.880232 E), located near the city of Hadera, Israel, on the easternmost Mediterranean Sea coast, is one of three coastal power stations found to attract sharks to their warm water effluent [1]. OR pumps seawater to cool its turbines and discharges the water back into the sea at approximately 8 °C above ambient temperature. The discharge creates a heated plume expanding a few kilometres along the coast, with a strong temperature gradient between the point of release and the ambient sea temperature (Figure 1).

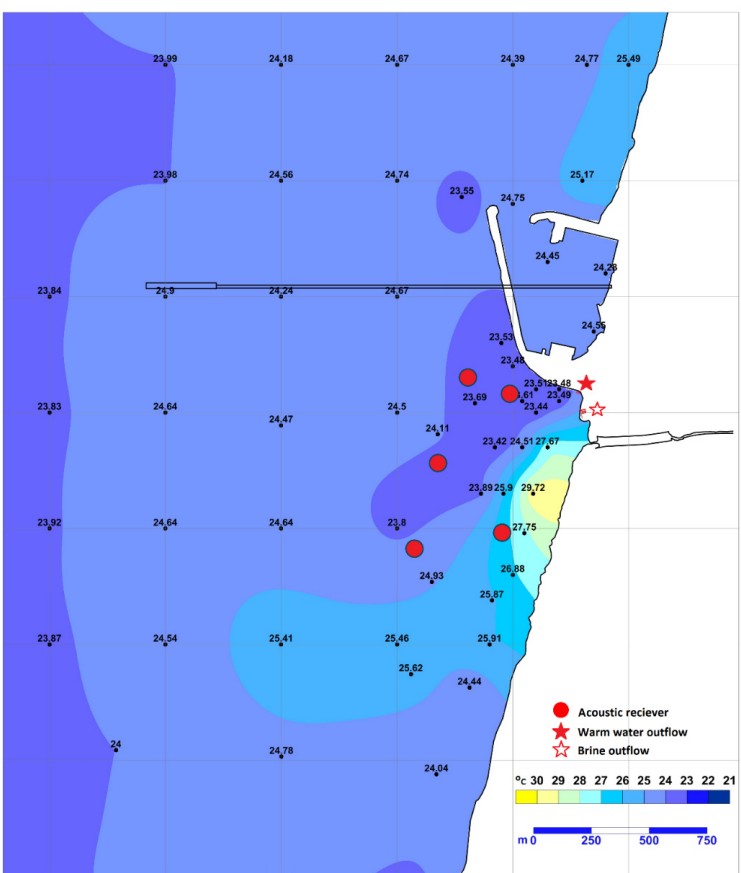

**Figure 1.** Map of the study site. Temperature is shown as measured by IEC staff on 22 May 2018, at 2 m depth. Circles (●) represent receiver locations; star (★) represents the discharge point for warm water and star with no fill (☆) for warm saline water. Adapted from the IEC monitoring report 2018.

In addition to OR, a desalination plant operates on-site and discharges its brine into the same effluent. As a result, the mixed water reaches the sea with a salinity about 3 PPT higher than the ambient seawater. The bottom depth at the discharge site ranges from 0 to 4 m in most places and reaches a maximum depth of 7.5 m in a certain area excavated by the discharge current.

*2.2. Shark Tagging and Receivers' Deployment*

Carcharhinid sharks were caught and tagged on-site in the warm effluent, between November 2017 and April 2018, using baited drumlines. The sharks were pulled close to the boat and were strapped around the tail base and behind the pectoral fins. Once secured, the sharks were measured, sexed (according to appearance or absence of claspers), and fitted with an external Floy tag in the dorsal fin. HP16 tags equipped with a depth sensor (Thelma Biotel, Trondheim, Norway; 69 kHz; delay range: 30–90 s; depth range: 0–51 m; resolution: 0.2 m; battery life: 90 months) were surgically implanted into the peritoneal cavity of 4 *C. plumbeus* and 9 *C. obscurus* sharks. Transmitters were set to nominally transmit every 60 s.

An acoustic receiver (VR2W, Vemco Ltd., Halifax, NS, Canada) was placed in the effluent on 15 January 2017, and four additional receivers (TBR 700, Thelma Biotel, Trondheim, Norway) were added on 7 March 2018.

*2.3. Data Analysis*

Data aggregation of a two-minute time frame was chosen to reduce the effect of the transition from one receiver to five and to even the number of detections. Mean depth (DM) was calculated for each aggregated data point. Detections from the first 24 h after tagging were discarded for each individual to eliminate tagging's effect on the movement analysis [9].

Day in the season (DIS) was used to describe the number of days starting 1 November for each season (an arbitrary date before the start of the tagging season), and a daily mean ambient seawater temperature (SWT) was calculated to check thermal changes. Time of day (TOD) was defined using the SUNCALC package [10], with the day divided into four time segments—Day, Night, Dawn, and Dusk—such that dawn was defined as the time between night's end (morning astronomical twilight start) to the end of the golden hour, approximately two hours later, and dusk was defined as between the beginning of the evening's golden hour and the beginning of night (dark enough for astronomical observations), which was approximately two hours later, as well. Lunar phase (LP) was added from the Lunar package. Total length (TL) represents the measured length of the shark on tagging. Data were then aggregated once more per Shark, DIS, and TOD. An aggregated data line based on three data points or less was removed, and the median (DM) value was chosen to describe the depth. Finally, a linear mixed model (LM, lmer function, Package lme4) was used to determine which factors affected the DM choice of the sharks. The model included interactions between the species and the TOD, and a random effect was included for individuals in order to control for possible dependences. A scale function was used to transform data to fit the same scale for all factors.

$$\text{DM Median} \sim \text{Species} \times \text{TOD} + \text{SWT} + \text{DIS} + \text{LP} + \text{TL} + (1 \mid \text{Shark})$$

Model selection was made by the Dredge function (Package MuMIn) with 5000 bootstrap resamples, showing 3 models with delta AIC < 2. Hedges G test was performed as post hoc for the model-chosen factors. Data analysis was performed in R (v. 1.8–12; R Foundation for Statistical Computing, Vienna, Austria).

**3. Results**

Sharks of the two species were caught along the tagging period and often in the same tagging event (three out of the four *C. plumbeus* sharks were tagged in the same event as a dusky shark. Table 1), providing proof of coexistence and mutual use of the heated area. All tagged *C. plumbeus* sharks were males, and all *C. obscurus* were females considerably larger than the *C. plumbeus* males (mean length ± SE: 298.2 ± 12.5 cm vs. 180 ± 4.5 cm respectively). These findings correspond with additional sharks caught and measured on site (Table A1, Appendix A) and with photographed observations showing mainly large female *C. obscurus* and smaller male *C. plumbeus* (unpublished data).

**Table 1.** Summary of biological and detection data for sharks tagged with depth sensors at the warm effluent of Orot Rabin (OR) power station, ordered by the tagging date. Detection rate stands for the number of detections per hour per receiver.

| Shark Serial | Species | Sex | TL (cm) | Detections | Tagging Date | Catch Time | Last Detected | Min Depth (m) | Max Depth (m) | Days Tracked | Detection Rate |
|---|---|---|---|---|---|---|---|---|---|---|---|
| CO 21 | *C. obscurus* | F | 289 | 318 | 27 November 2017 | 10:30 | 11 March 2018 | 1 | 6.8 | 105 | 3.2 |
| CO 23 | *C. obscurus* | F | 276 | 737 | 12 December 2017 | 14:34 | 24 April 2018 | 1 | 15 | 134 | 6.3 |
| CO 22 | *C. obscurus* | F | 315 | 482 | 27 December 2017 | 7:17 | 2 April 2018 | 1 | 7 | 97 | 6.1 |
| CO 14 | *C. obscurus* | F | 355 | 424 | 27 December 2017 | 10:43 | 13 March 2018 | 1 | 7.4 | 77 | 7.4 |
| CO 20 | *C. obscurus* | F | 300 | 267 | 2 January 2018 | 13:00 | 8 May 2018 | 0 | 9.2 | 127 | 3.1 |
| CO 26 | *C. obscurus* | F | 275 | 1051 | 5 February 2018 | NA | 22 April 2018 | 1 | 17.6 | 77 | 8.2 |
| CP 15 | *C. plumbeus* | M | 169 | 17117 | 12 March 2018 | 13:00 | 14 May 2018 | 0.6 | 13.8 | 64 | 53.5 |
| CO 25 | *C. obscurus* | F | 280 | 63 | 12 March 2018 | 13:00 | 23 March 2018 | 1.6 | 7.4 | 12 | 1.1 |
| CP 10 | *C. plumbeus* | M | 191 | 17231 | 14 March 2018 | 10:55 | 10 May 2018 | 0.6 | 17 | 58 | 59.4 |
| CO 11 | *C. obscurus* | F | 294 | 969 | 28 March 2018 | 8:52 | 27 April 2018 | 1 | 38.6 | 31 | 6.3 |
| CP 17 | *C. plumbeus* | M | 180 | 4706 | 28 March 2018 | 11:59 | 14 May 2018 | 0 | 11.2 | 48 | 19.6 |
| CO 12 | *C. obscurus* | F | 300 | 1895 | 2 April 2018 | 11:33 | 2 June 2018 | 0.8 | 7.6 | 62 | 6.1 |
| CP 27 | *C. plumbeus* | M | 180 | 4348 | 2 April 2018 | 13:49 | 21 April 2018 | 1.4 | 10.8 | 20 | 43.5 |

A linear mixed-model analysis found movement in DM best explained by three top models, which included the species, time of day (TOD), and day in the season (DIS). The model did not find the ambient temperature, lunar phase, or the shark's total length to significantly affect the DM. Residuals distribution for the model appears in Figure A2.

The Akaike information criterion (AIC) was similar to the first 3 models (delta < 2), and all 3 models were able to account for 58% of the variance (Table 2).

**Table 2.** Model selection results only include models with $\Delta$AIC < 2. DM represents median depth, TOD represents the category time of day, DIS represents day in season, and Shark represents an individual shark.

| Model Formula | AICc | ΔAICc | df | Log Likelihood | adj$R^2$ |
|---|---|---|---|---|---|
| DM ~ Species + TOD + (1 \| Shark) | 1562.2 | 0 | 7 | −774.010 | 0.579 |
| DM ~ Species + TOD + DIS + (1 \| Shark) | 1563.1 | 0.9 | 8 | −773.453 | 0.583 |
| DM ~ Species × TOD + (1 \| Shark) | 1564.1 | 1.9 | 10 | −771.883 | 0.585 |

*C. plumbeus* were deeper than *C. obscurus* at all times of the day, with a mean difference of 1.5 m during the day and at night (Figure 2). In crepuscular times, this number changes towards a higher number (1.8 m) at dawn and a lower number (1.26 m) at dusk, suggesting *C. plumbeus* might start the movement earlier than *C. obscurus*, thus creating a bigger gap in the morning and a smaller one going back up at night.

This result was repeated when comparing DM at the different TOD within each species. Compared to DM at night, *C. obscurus* ventured 1.39 m deeper during the day (there was no significant difference between DM at night and the transient times), whereas *C. plumbeus* changed their DM significantly early at dawn and continued moving 2 m deeper for the day. DM at dusk was not significantly different from the night (Table 3).

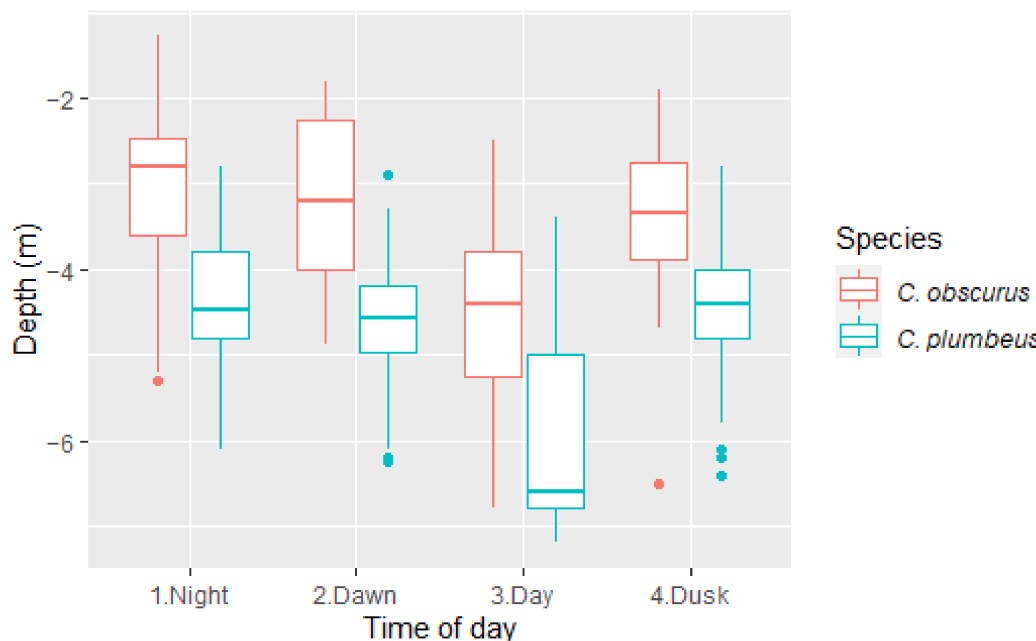

**Figure 2.** Median depth by the time of day for *C. obscurus* and *C. plumbeus*. The upper and lower limits of boxes represent the 75th and 25th percentiles, respectively. Horizontal lines represent the median value.

**Table 3.** Unpaired Hedges' g test among species and time of day groups.

| Species | Test | Difference (m) | 95 CL | Sig |
|---|---|---|---|---|
| *C. obscurus* | Dawn (*n* = 21)—Night (*n* = 95) | 0.0689 | [−0.382; 0.549] | − |
| *C. obscurus* | Day (*n* = 87)—Night (*n* = 95) | 1.39 | [1.05; 1.73] | + |
| *C. plumbeus* | Dawn (*n* = 66)—Night (*n* = 125) | 0.351 | [0.0516; 0.639] | + |
| *C. plumbeus* | Day (*n* = 116)—Night (*n* = 125) | 2 | [1.67; 2.34] | + |
| *C. plumbeus* | Dusk (*n* = 82)—Night (*n* = 125) | 0.166 | [−0.122; 0.434] | − |
| **TOD** | **Test** | **Difference (m)** | **95 CL** | **Sig** |
| Night | *C. plumbeus* (*n* = 125)—*C. obscurus* (*n* = 95) | 1.48 | [1.11; 1.84] | + |
| Dawn | *C. plumbeus* (*n* = 66)—*C. obscurus* (*n* = 21) | 1.80 | [1.15; 2.42] | + |
| Day | *C. plumbeus* (*n* = 116)—*C. obscurus* (*n* = 87) | 1.51 | [1.19; 1.85] | + |
| Dusk | *C. plumbeus* (*n* = 82)—*C. obscurus* (*n* = 24) | 1.26 | [0.556; 1.87] | + |

## 4. Discussion

The aggregation of sharks at OR's effluent provides a unique opportunity to examine how human development causes a change in the movement and behaviour of certain shark species, as well as the behavioural adaptations of the sharks to the new conditions in terms of competition and use of space. In this study, we describe this aggregation behaviour, and the vertical movement patterns within it, at an individual level, as well as offer a possible explanation for the observed coexistence between these species at the site.

Clear and constant diel vertical movement was found for both species at the site. All sharks swam in the upper water column at night and ventured deeper during the day, although the shift of DM between day and night was characterized by a seemingly minor difference for sharks of that size (i.e., a change of no more than 2 m for 2–3.5 m long sharks). A distinct difference in utilised DM was found between the species, showing *C. plumbeus* swam deeper than *C. obscurus*, displaying spatial partitioning of the species. Moreover, the only place within the heated area to reach a depth greater than 5 m is underneath the

discharge current, where *C. plumbeus* sharks have been documented repeatedly (Figure 3). The utilised DM (for each species) was not related to the ambient SWT, the lunar phase, or the individual size of the sharks, suggesting a species-specific spatial partitioning at the study site. These results are further reinforced by the swimming profile recorded by an archival tag attached to one of the *C. plumbeus* sharks (Figure A1).

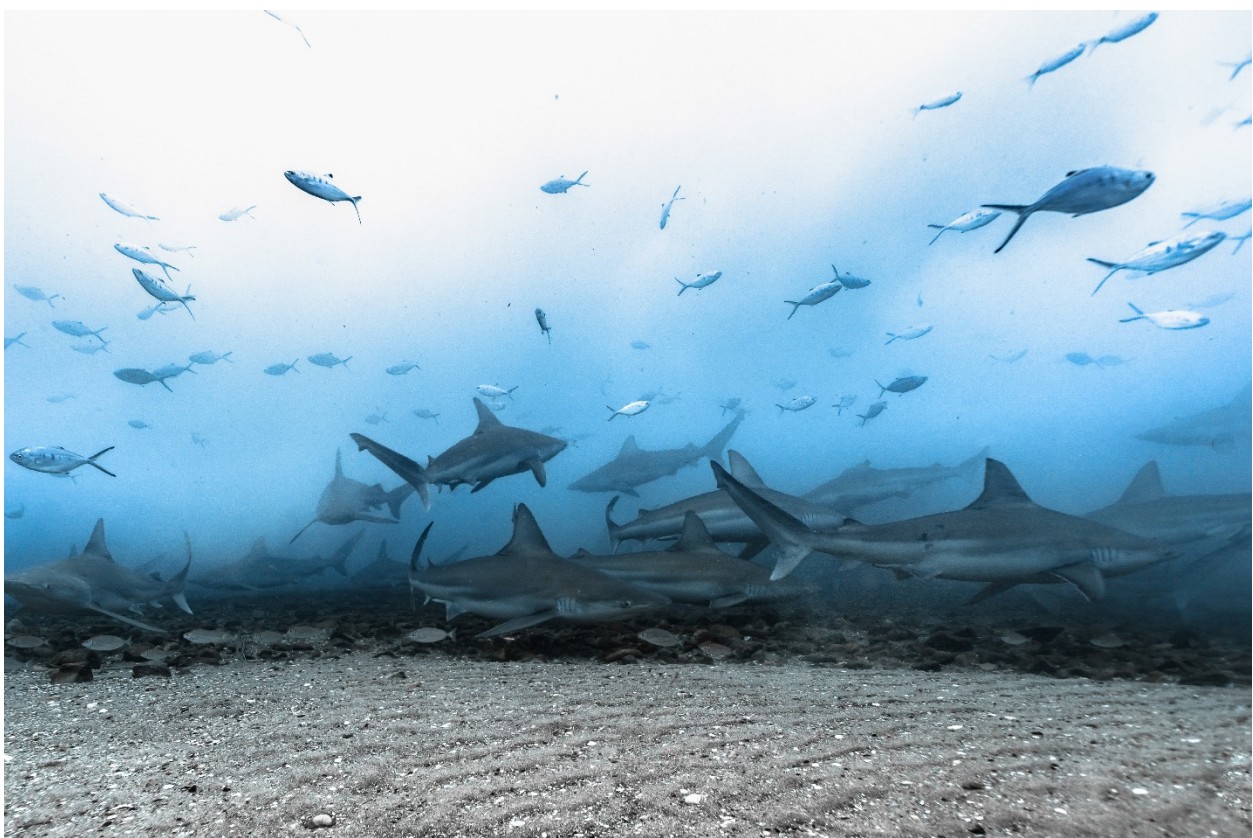

**Figure 3.** *Carcharhinus plumbeus* swimming under the current at 7 m. (The photo was reprinted with permission from Ilan Elgrably).

The idea of spatial partitioning is further supported by the order of magnitude that was found in the difference in the detection rate of *C. plumbeus* (Table 1), suggesting different utilization of the space by *C. obscurus* and *C. plumbeus* at the study site. The number of detections, however, may be affected by the acoustic noise the artificial current causes in shallower waters.

In this study, the scale of DM variation was very small (due to the nature of the study site), as was the difference in sizes within each species. All *C. obscurus* individuals were considerably larger than the *C. plumbeus* individuals, and therefore, it is impossible to fully determine whether the daily changes in spatial occupation were due to individual size, species, or sex. Here we observed a few dozen sharks of each species coexisting in "close quarters", seemingly facilitated by a daily "shift-change" in terms of time and DM locations. Recently, temporal shifts have been shown between sharks of different species in Tampa, Florida, demonstrating robust temporal partitioning of foraging times [7]. This might also be the case here, with *C. plumbeus* waiting their turn to feed.

Diel movement may be driven by prey behaviour [8,11]. *C. plumbeus* and *C. obscurus* mainly feed on teleost fish and cephalopods [12–15] and are considered to be at the same trophic level (4.1 for *C. plumbeus* and 4.2 for *C. obscurus*, Cortés, 1999), but size differences between the species at the study site could be driving differences in feeding preferences, as has been suggested for other species [16–18].

Inter-species competition can also explain the difference between the movements of the sharks. The larger *C. obscurus* spent time at the site freely during the day, while the smaller *C. plumbeus* entered the "preferred" depth at night when *C. obscurus* individuals were not there. The slight change in the timing of the transition between deep and shallow supports the theory that one species "responds" to the movement of the other species.

The idea of division in depth utilization according to sharks' size has been suggested by [19], where smaller sixgill sharks (*Hexanchus griseus*) used shallower sites than larger individuals; however, this was only observed in individuals of the same species. In this study, the total length of individual sharks was not significant, but it could be overshadowed compared to the size variation between the two species.

Salinity has also been found to be a driver in shark movement. Reference [5] found two species of river sharks segregated spatially along a salinity gradient. This possibility should be further explored at the study site in terms of salinity tolerance and/or preference for both species and whether it plays a part in the species' depth distribution.

The unique circumstances provided by the shark aggregations at OR allow us to examine changes in DM on a scale that is rarely possible. It seems that diel vertical movement was maintained, even though functionally, the differences in depth are considered minor compared to the vertical movement reported for sharks of the same species in different areas. These findings may suggest that vertical diel movement is an inherently basic behaviour in sharks of these species and is maintained, even in cases when it is not essential.

**Author Contributions:** Conceptualization, A.B. and D.T.; data curation, A.B., A.S., E.B., Z.Z.S. and S.M.; formal analysis, A.B.; investigation, A.B.; methodology, A.B. and A.S.; supervision, D.T.; visualization, A.B.; writing—original draft, A.B.; writing—review & editing, A.S. and S.M. All authors have read and agreed to the published version of the manuscript.

**Funding:** This study was funded by the Morris Kahn Marine Research Station, Department of Marine Biology, Leon H. Charney School of Marine Sciences, University of Haifa, Israel.

**Institutional Review Board Statement:** Shark tagging was conducted under permit numbers 2017/41714 and 2018/42027, issued by The Israeli Nature and Parks Authority (INPA), and according to European ecological standards.

**Informed Consent Statement:** Not applicable.

**Data Availability Statement:** Data are available from the authors upon reasonable request and with permission of the "Top Predator Lab" at Morris Kahn Marine Research Station, Department of Marine Biology, Leon H. Charney School of Marine Sciences, University of Haifa, Israel.

**Acknowledgments:** We thank Ilan Elgrably for his photograph (Figure 3), Ran Golan for fruitful discussions, and Kfir Avramzon the maritime lab manager in the engineering projects division, IEC, Israel.

**Conflicts of Interest:** The authors declare no conflict of interest.

## Appendix A

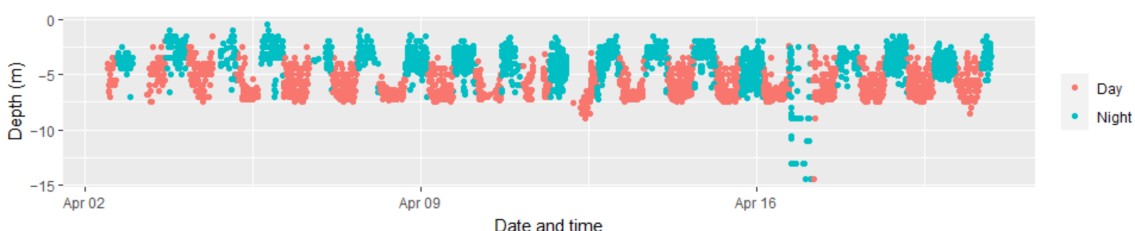

**Figure A1.** Depth data as was recorded in an archival tag of a *Carcharhinus plumbeus* male. Points are coloured according to the time of day (day in red, night in blue).

**Table A1.** Size measurements of untagged sharks captured within the study site between 2016–2017.

| Species | Catch Date | TL (cm) | Sex |
|---|---|---|---|
| *C. obscurus* | 25 February 2016 | 322 | Female |
| *C. obscurus* | 25 February 2016 | 328 | Female |
| *C. obscurus* | 23 March 2016 | 309 | Female |
| *C. obscurus* | 23 March 2016 | 325 | Female |
| *C. obscurus* | 23 March 2016 | 299 | Female |
| *C. obscurus* | 17 January 2017 | 200 | Female |
| *C. obscurus* | 20 February 2017 | 250 | Female |
| *C. obscurus* | 21 February 2017 | 290 | Female |
| *C. obscurus* | 23 February 2017 | 280 | Female |
| *C. obscurus* | 6 March 2017 | 280 | Female |
| *C. obscurus* | 8 March 2017 | 390 | Female |
| *C. obscurus* | 28 March 2017 | 320 | Female |
| *C. obscurus* | 19 December 2017 | 283 | Female |
| *C. obscurus* | 9 January 2018 | 303 | Female |
| *C. plumbeus* | 8 March 2017 | 170 | Male |
| *C. plumbeus* | 23 February 2017 | 177 | Male |
| *C. plumbeus* | 6 April 2017 | 198 | Male |
| *C. plumbeus* | 6 April 2017 | 179 | Male |
| *C. plumbeus* | 1 May 2018 | 178 | Male |

**Figure A2.** Residuals distribution for the LM model.

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
