# Peer review of "Depth Partitioning and Diel Movement of Two Large Carcharhinid Sharks in Extremely Shallow Waters"

_fishes, doi:10.3390/fishes8020085_

Round 1
Reviewer 1 Report
-English language quality must be improved throughout the manuscript, e.g. line 32: it must be “have aggregated” instead of “aggregate”.
-Citations and references must be revised according to the journal instructions.
-Be consistent with using “m” and “meters” (although “meters” is regularly used throughout the manuscript, “m” is used, e.g., twice in line 131. Furthermore, there should be no dot behind cm (written twice in line 119).
-Lines 11–12: species authorities must be indicated upon first mentioning of a species name.
-Line 12: write “plumbeus” instead of “Plumbeus”
-Lines 34–35: maximum sizes are not correct, see Weigmann (2016: Annotated checklist of the living sharks, batoids and chimaeras (Chondrichthyes) of the world, with a focus on biogeographical diversity. Journal of Fish Biology 88(3), 837–1037. https://doi.org/10.1111/jfb.12874) for maximum total length values.
-2.2. Shark Tagging and receivers’ deployment (it must be “tagging” instead of “Tagging”): as sharks were fitted with external tags and, particularly, had surgical implantation of HP16 tags, an ethical justification is needed.
-Lines 87: the abbreviation “SD” for “Day in the Season” (should be “season” instead of “Season” is not optimal as “SD” is usually used for abbreviating standard deviation. Therefore, I would recommend using a different abbreviation for day in the season.
-Line 93: it must be “Lunar phase” instead of “Lunarphase”.
-Line 104: “LM” must be explained.
-Lines 119–121: why are these data not included in the present manuscript? I encourage inclusion of these data particularly considering that the informative value and amount of data of the present manuscript is somewhat limited and only based on a rather small number of sharks.
-Line 128: rephrase the table header, e.g. “Model selection results only including models with ∆AIC<2.”
-Line 140: caption to Figure 2: abbreviations should be explained so that figure captions can be understood independent from the manuscript body.
-Line 151: use n-dash for ranges instead of minus sign.
-Line 159: although a photograph taken by a non-author of the manuscript is used, the photographer is not acknowledged in the manuscript. An acknowledgment should be added in an Acknowledgements section. Similarly, an acknowledgement is also missing for the photographer of the photograph shown in the appendix.
-Line 160: write “Carcharhinus” instead of “C.” and write Carcharhinus plumbeus in italics. The genus name should also be written in full at the beginning of line 219.
Reviewer 2 Report
This is a good paper with a modest amount of data. I have recommended that two items in the Appendix be added to give the article more scientific 'meat'. With a little revision this paper, which is an interesting and a significant one, could be made much better. Hence my rating the quality of presentation is average and not high. It should be published in Fishes upon revision. I have tried to give the authors some guidance on how to make the paper better. My comments are given below. I am recommending major revision based on the improper use of parametric indicators for normal distributions of data in Figure 2 when some distributions seem non-normal. Also, more information should be given on the transmitters and on the range of transmitter detection.
General
(1) More information should be provided about the HP16 tags. What is their size, signal strength, pseudo interval between identification pulse bursts? What is the resolution (smallest depth differences) and accuracy, nearness to true temperature), and precision (similarity of repeated measurements). These parameters are important as the sharks change depth in small increments.
(2) The range of detection should be provided, ideally based upon a range test, as provided by most papers using coded acoustical tags and autonomous monitors. Note that these were conducted in the Klimley et al. (2022) paper.
(3) The authors should consult a statistician to make sure the statistics are appropriate. The horizontal lines on the histogram bars in Figure 2 indicate means. However, their distributions are not always normal (see the C. Plumbeus bar for Day, where the horizontal line is near the -SE). Plotting histograms with medians, 75th percentiles, and ranges would be more appropriate. The Hedges’ g test should be applicable to non-normal data, as are the Chi Square Tests. It would be good to include the table at Line 213 in the body of the paper, as statistical significance is important.
(4) The graph on line 214 shows a consistent change in depth of a sandbar male. It should be included in the body of the manuscript.
(5) The picture of the shark swimming above another seems not needed – just a statement that it occurs. The picture of the sandbar sharks swimming under the current is worthwhile and should be kept in the manuscript.
Specifics
Line 11: Replace “haves” with “have”.
Line 20: Replace “depts” with “depths”, remove “at”
Line 91-92. How was dawn and dusk defined, civil, nautical twilight, astronomical twilight?
See:
Qayum, H.A., A.P. Klimley, J.E. Richert, and R. Newton. 2006. Broad-band versus narrow-band irradiance for estimating latitude by archival tags. Marine Biology, 151: 467-481.
“Geometrical sunset occurs when the ‘‘upper limb’’ or top of the sun’s disk coincides with the horizon and its waist is 0.27_ below the horizon (Fig. 1b). The observed sunset occurs later, when the waist of the sun is 0.82_ below the horizon, because light bends (refracts) 0.55_ around the earth’s atmosphere, resulting in an apparent image of the sun after the sun has passed below the horizon (Nautical Almanac Office 1997). There is a period of time after sunset and before sunrise, twilight, when irradiance originates only from the upper atmosphere, which receives direct sunlight and reflects part of it toward the surface of the earth (US Naval Observatory 2006). First is civil twilight, which terminates when the center of the solar disk is 6° below the horizon (or a = –6.0°). During this period, the illumination is adequate, under ideal weather conditions, to distinguish terrestrial objects from the dark background, see the horizon, and observe the brightest stars in the absence of the moon. Nautical twilight follows and continues until a = –12° and during this period the vague outlines of objects can be seen, but not the horizon. Astronomical twilight follows and expires at a = –18°. During this period, the tangent with the horizon. Observed sunset occurs when waist of the sun is 0.82° below the horizon, because light bends (refracts) 0.55° around the earth’s atmosphere, resulting in an apparent image of the sun after the sun has actually passed below the horizon.”
The reviewer assumes the authors use nautical twilight in their study.
Line 139:
Line 142. The selection of the warm effluent from a power is not unique to sharks. Marine turtles show the same behavior.
See:
Madrak, S.V., R.L. Lewiston, T. Eguchi, A.P. Klimley, and J.A. Seminoff. 2022. Effects of ambient temperature on dive behavior of East Pacific green turtles before and after a power plant closure. Marine Ecology Progress Series, 683: 157-168; doi.org/10.3354/meps13940.
Line 187: This sentence is incomplete, ending with “that” with no period.
Round 2
Reviewer 1 Report
The manuscript has been improved significantly but there are still few minor issues that need to be addressed before it can be accepted for publication:
-Species authorities have still not been added upon first mentioning of a species.
-The table in Appendix 2 is informative, but the unit used for the TL measurements needs to be added (likely [m]). Also the reference in the text seems to be incorrect (the manuscript says "These findings correspond with additional sharks caught and measured on site (Appendix 3)").
-In line 38 a there is a crossed out "a".
-In line 59, use comma and blank space instead of minus.
-In line 84, abbreviation "DM" needs to be explained.
-In line 100, second mentioning of "Dusk" should be "dusk" as it refers to the generally valid word dusk and no specific package name.
-In line 187, "c plumbeus" should read "C. plumbeus" and "c. obscurus" should read "C. obscurus", both written in italics.
-Tables: Table 1 should be referred to in the text earlier already and Table 3 is not yet referred to in the text at all. Reference to Table 2 should be "(Table 2)" instead of "(table 2)". In the header to Table 1, abbreviation "OR" should be explained so that the table can be understood independently from the manuscript text. Also, abbreviations used in Tables 2 und 3 should be explained in the respective table header.
-Figure 2: species names must be written in italics in the figure legend.
Author Response
-Species authorities have still not been added upon first mentioning of a species. added
-The table in Appendix 2 is informative, but the unit used for the TL measurements needs to be added (likely [m]). added and changed to cm
Also the reference in the text seems to be incorrect (the manuscript says "These findings correspond with additional sharks caught and measured on site (Appendix 3)"). fixed
-In line 38 a there is a crossed out "a". deleted
-In line 59, use comma and blank space instead of minus. changed
-In line 84, abbreviation "DM" needs to be explained. rephrased
-In line 100, second mentioning of "Dusk" should be "dusk" as it refers to the generally valid word dusk and no specific package name. fixed
-In line 187, "c plumbeus" should read "C. plumbeus" and "c. obscurus" should read "C. obscurus", both written in italics. fixed
-Tables: Table 1 should be referred to in the text earlier already and Table 3 is not yet referred to in the text at all. Reference to Table 2 should be "(Table 2)" instead of "(table 2)". All three tables were adjusted.
In the header to Table 1, abbreviation "OR" should be explained so that the table can be understood independently from the manuscript text. Also, abbreviations used in Tables 2 und 3 should be explained in the respective table header. rephrased
-Figure 2: species names must be written in italics in the figure legend. fixed
Reviewer 2 Report
I asked the authors to define dawn and dusk. There are technical definitions, i.e., civil twilight, nautical twilight, and astronomical twilight. The authors use a program and define it as "the golden hours between day and night (NightEnd to GoldenHourEnd and GoldenHour to Night). What is this?
Note that I wrote in my initial review:
Line 91-92. How was dawn and dusk defined, civil, nautical twilight, astronomical twilight?
See:
Qayum, H.A., A.P. Klimley, J.E. Richert, and R. Newton. 2006. Broad-band versus narrow-band irradiance for estimating latitude by archival tags. Marine Biology, 151: 467-481.
“Geometrical sunset occurs when the ‘‘upper limb’’ or top of the sun’s disk coincides with the horizon and its waist is 0.27_ below the horizon (Fig. 1b). The observed sunset occurs later, when the waist of the sun is 0.82_ below the horizon, because light bends (refracts) 0.55_ around the earth’s atmosphere, resulting in an apparent image of the sun after the sun has passed below the horizon (Nautical Almanac Office 1997). There is a period of time after sunset and before sunrise, twilight, when irradiance originates only from the upper atmosphere, which receives direct sunlight and reflects part of it toward the surface of the earth (US Naval Observatory 2006). First is civil twilight, which terminates when the center of the solar disk is 6° below the horizon (or a = –6.0°). During this period, the illumination is adequate, under ideal weather conditions, to distinguish terrestrial objects from the dark background, see the horizon, and observe the brightest stars in the absence of the moon. Nautical twilight follows and continues until a = –12° and during this period the vague outlines of objects can be seen, but not the horizon. Astronomical twilight follows and expires at a = –18°. During this period, the tangent with the horizon. Observed sunset occurs when waist of the sun is 0.82° below the horizon, because light bends (refracts) 0.55° around the earth’s atmosphere, resulting in an apparent image of the sun after the sun has actually passed below the horizon.”
The reviewer assumes the authors use nautical twilight in their study.
This should be clarified before the article is accepted for publication.
Author Response
I have rephrased, I hope it is clear now.